# An Energy-Efficient Routing Algorithm for WSNs Using Fuzzy Logic

**DOI:** 10.3390/s23198074

**Published:** 2023-09-25

**Authors:** Preetha R. Rao, Amruta Lipare, Damodar Reddy Edla, Saidi Reddy Parne

**Affiliations:** 1National Institute of Technology Goa, Ponda 403401, Goa, India; preetharao21@gmail.com (P.R.R.); dr.reddy@nitgoa.ac.in (D.R.E.); psreddy@nitgoa.ac.in (S.R.P.); 2Department of CSE, Indian Institute of Information Technology Pune, Pune 411041, Maharashtra, India

**Keywords:** wireless sensor networks, base station, cluster head, energy efficiency, fuzzy logic, load balancing, routing

## Abstract

Battery replacement or recharging is essential for sensor nodes because they are typically powered by batteries in wireless sensor network (WSN) applications. Therefore, creating an energy-efficient data transfer technique is required. The base station (BS) receives data from one sensor node and routes the data to another sensor node. As a result, an energy-efficient routing algorithm using fuzzy logic (EERF) represents a novel approach that is suggested in this study. One of the reasoning techniques utilized in scenarios where there is a lot of ambiguity is fuzzy logic. The remaining energy, the distance between the sensor node and the base station, and the total number of connected sensor nodes are all inputs given to the fuzzy system of the proposed EERF algorithm. The proposed EERF is contrasted with the current systems, like the energy-aware unequal clustering using fuzzy logic (EAUCF) and distributed unequal clustering using fuzzy logic (DUCF) algorithms, in terms of evaluation criteria, including energy consumption, the number of active sensor nodes for each round in the network, and network stability. EAUCF and DUCF were outperformed by EERF.

## 1. Introduction

Wireless channels connect the sensor nodes in a wireless sensor network, which are then placed throughout the region being monitored. Sensor node placement may be random or static [1,2]. Depending on the applications, some mobile devices can be connected to the sensor nodes. Sensor nodes keep an eye on the area and measure environmental factors like temperature and humidity. The base station (BS) receives the collection of data and transmits it. In order to obtain information from the targeted region, the BS is connected to the internet to deliver notifications about the target area [3,4]. The transmission of data by sensor nodes uses a lot of energy. In order to decrease energy consumption, clustering and routing are used. The grouping of sensor nodes is one of the efficient techniques used for utilizing energy effectively. A sensor node in the clustering method leads to a specific cluster, known as the cluster head (CH). The CH from each cluster of the network gathers information from every sensor node and then transmits it to the BS. The application of the routing mechanism prevents direct data transmission from CHs to the BS [5]. Figure 1 shows both the routing path between CHs and the WSN with clusters.

The network’s lifespan is significantly increased through routing. It is difficult to build a routing path for a network of sensor nodes because they have little computational power, short transmission ranges, and limited battery life. Since the next hop does not take the energy of the sensor node into account, using minimum hop might not be the best option. The hop will continue in this case until the sensor node fails or dies. As a result, the network develops an energy hole and becomes dispersed. A protocol that guarantees uniform energy consumption by sensor nodes is necessary to extend the network’s lifespan. There are three basic types of routing protocols: location-based routing, hierarchical routing, and data-centric flat routing. Because sensor nodes and cluster heads aggregate data during hierarchical routing (cluster-based routing), fewer packets are sent throughout the network, which reduces energy consumption. Cluster-based routing utilizes multi-hop communication and aids network scalability [6].

As energy is a crucial factor in WSNs, designing energy-efficient algorithms to communicate is necessary. As the routing distance increases, the energy consumption of the network increases. Therefore, it is also essential to route the data sensed by the sensor nodes to the base station through an efficient path. In this paper, the fuzzy system is used to route the data using an energy-efficient routing path. The existing EERF supports multi-hop routing by generating the least distant and most energy-efficient routing path, whereas in energy-aware unequal clustering using fuzzy logic (EAUCF) and distributed unequal clustering using fuzzy logic (DUCF), clustering is preferred for the communication of the data.

Making decisions that follow human logic is an essential application of fuzzy logic. Fuzzy logic makes decisions depending on the result by combining different features on pre-established principles. Fuzzifiers, fuzzy rules, fuzzy inference systems, and defuzzifiers are some of fuzzy logic’s key elements. Fuzzifier creates a fuzzy set from a clean input. Each set is given a degree of membership as well. The connections between linguistic input variables and fuzzy output variables are made by fuzzy rules. The fuzzy inference system uses rules to process values that have been fuzzified. By utilizing a defuzzifier, the fuzzy outputs are provided as a solution in terms of crisp values. A fuzzy logic-based routing algorithm has been suggested in this study. The development of the routing path and router selection are carried out in accordance with the fuzzy output values. Section 2 of the proposal describes the work that went into this. In Section 3, the proposed routing algorithm has been described. The experimental findings are discussed in Section 4. At the end, Section 5 brings the paper to a conclusion.

## 2. Related Work

### 2.1. Literature Survey

There are various clustering strategies for WSNs in the literature. One of the traditional techniques used for CH selection and cluster formation is low energy adaptive clustering hierarchy (LEACH) [7]. In LEACH, where ‘*p*’ is the predetermined number of rounds, the CHs are chosen with a probability of 1/p at each round. Following *p* rounds, some sensor nodes are selected as CHs. Consequently, after a predetermined number of rounds, a sensor node that has a high amount of energy may be selected as a CH. As a result, the sensor dies more quickly and loses power more quickly. Dead sensor nodes, in this scenario, prevent participation in the tournament; hence, there will be no transmission in that round. This problem has been solved by researchers by enhancing LEACH. By using fuzzy logic, Kim et al. suggested a novel method for cluster head election (CHEF) [8]. The remaining energy and distance of the sensor node from the BS serve as input variables for the fuzzy system in CHEF.

Similar to this, Hagci et al. created the EAUCF, a novel uneven clustering technique [9]. A constant threshold (*T*) value in the EAUCF is predefined and ranges from 0 to 1. Each of the sensor nodes in the network generates a random number between 0 and 1. The generated value contrasts with the threshold value *T*. Speculative CHs are considered to be sensor nodes with random values below *T*. The CHs serve as the input for the fuzzy inference models. ‘Residual Energy’ and ‘Distance to BS’ are provided as input values for the fuzzy system. The proposed fuzzy system generates ‘Competition Radius’ as an output value for the proposed EAUCF. The value of the competition radius of CHs is used to determine the final CHs.

New fuzzy algorithm DUCFs were developed by the authors of [10]. Here, residual energy, distance to BS, and node degree—a new input parameter for fuzzy inference systems—are taken into account. The node degree is the quantity of sensor nodes that are within its communication range. Clusters are created based on the fuzzy system’s output values or the “Chance” and “Size” of the CHs.

LEACH, CHEF, and EAUCF have the drawback of not taking into account the load of the data packets that must be traversed in order to choose the final CHs. The researchers behind this study employed the same clustering technique as DUCF [10] and proposed a routing algorithm that makes use of a fuzzy rule-based system. The outputs of the fuzzy system select the appropriate routers and build a routing path.

Uneven clustering using multi-hop transmission based on fuzzy logic is advised in order to effectively regulate energy usage among the nodes. Here, based on competition radius, the protocol creates uneven groups, and the cluster leader is chosen using fuzzy logic. The node concentration, node distance from the sink, and remaining energy are the input variables in this case. In comparison to the current EAUCF and LEACH algorithms, for parameters like energy efficiency and network lifetime, this protocol performs better [11]. Within the communication range of a current forwarder or a source sensor, it has been shown that single-hop forwarding methods utilize less energy than multi-hop forwarding schemes. The authors used the social welfare function to predict the inequality of neighbors’ residual energy after selecting a number of next-hop nodes. Based on energy disparity, a technique was developed to assess the degree of energy balance. How close a node is to the shortest path, how close it is to the sink, and how much energy balance there is are all parameters that are provided to the fuzzy logic system. A multiparameter, fuzzy routing decision is made by using a fuzzy logic-based energy-optimized routing algorithm. In comparison to other algorithms of a similar nature, the algorithm successfully extends the network lifetime and achieves both energy efficiency and energy balance, according to simulation findings [12].

The lifespan of a WSN is significantly impacted by the energy consumption of the sensor nodes. Energy dissipation among sensor nodes as well as energy-efficient operation are both necessary for extending a network’s lifespan. On the other hand, fluctuations in sensor activity over time and space lead to an energy imbalance throughout the network. In order to increase the network’s longevity, routing algorithms should properly balance energy efficiency and consumption. The distributed energy-aware fuzzy logic-based routing algorithm (DEFL) that the authors suggest handles both energy efficiency and energy balance at once. With the purpose of calculating the shortest path, the design converts the network condition into the corresponding cost values using the necessary energy measurements. In order to add human logic to the mapping, the authors use a fuzzy logic technique [13].

However, by choosing the most effective relay node in communication with the sink node, the authors of [14] presented fuzzy-based relay node selection and energy-efficient routing (FRNSEER). Fuzzy rules were applied to choose the sink node, with the active relay nodes chosen as a result. The choice of an active relay node aids in determining the best energy and utility factor for the transmission process. Sensor hubs, which use a timetable that uses less energy and anticipates the functionality of neighboring or working nodes, were utilized to improve communication between the sink node and the relay node. As a part of the performance analysis, the suggested mechanism outperforms existing algorithms like fuzzy-based hyper round policy and neural network-based localization scheme (NNBLS) in terms of packet rate while using less energy. The input values of the clustering and routing algorithms are associated in [15] by using rule-based fuzzy logic. The fuzzy system makes judgments based on the number of nodes in the communication range, the remaining energy of the sensor nodes, and the distance of the sensor nodes from the BS. The crisp values of the input variables are converted to various fuzzy values. By using the centroid defuzzification method, the fuzzy output values are transformed into crisp values. The output values are taken into consideration while choosing the cluster heads (CHs) and routers. As a result, according to the CHs’ ability to handle a certain load, the sensor nodes are allocated to the appropriate CHs from the network. Further, the capacity of the routers determines the routing path.

In [16], the cluster is created using the particle swarm optimization (PSO) method, and a fuzzy-based energy-efficient routing protocol (E-FEERP) is proposed to transmit data from the cluster head to the BS in the best possible way. This protocol takes into account the average distance of the SN from the BS, node density, energy, and communication quality.

A fuzzy-based approach combined with the grey wolf optimization algorithm was suggested in [17], which assists cluster formation in finding an effective and optimal solution for selecting the aggregation points using the cluster heads and figuring out the best optimal path for transmitting data to the network base station.

The hierarchical fuzzy multi-criteria-clustering and bio-inspired energy-efficient routing (FMCB-ER) protocol is presented in [18] to improve the lifespan of networks and, consequently, the operating time of WSN-based applications. In this method, the robust clusters are formed using a grid-based clustering methodology.

### 2.2. Energy Model

The network’s overall energy usage is determined by using the radio energy model from [19]. Each sensor node in a WSN requires energy to carry out various clustering tasks, send and receive packets from other sensor nodes, and do other tasks. Equation (Equation 1) describes how much energy is used by a sensor node to transmit a data packet with *l* bits over a length of *d* meters.
(1)$$E_{T}\left(l,d\right)=\left\{\begin{array}{cc} l*E_{elec}+l*\epsilon_{fs}*d^{2}, & d<d_{0}\\ l*E_{elec}+l*\epsilon_{mp}*d^{4}, & d\geq d_{0} \end{array}\right.$$
where the energy needed for data transmission in the free-space channel and the multi-path communication channel, respectively, are $\epsilon_{fs}$ and $\epsilon_{mp}$. The transmission distance threshold value is $d_{0}$. The total amount of energy used by a sensor node’s electronic circuitry is measured as $E_{elec}$. Equation (Equation 2) formulates the energy needed for receiving data of *l* bits in length.
(2)$$E_{R}\left(l\right)=l*E_{elec}$$

## 3. Proposed Work

The clusters in this work are constructed in accordance with DUCF [10]. In order to choose routers from CHs and create a routing path, an algorithm is suggested. From CHs to routers, the data are sent in the direction of the BS. The fuzzy routing process comprises three stages: the fuzzy router competition phase, the router selection phase, and the route formation phase. These phases are described in the sections below.

### 3.1. Fuzzy Router Competition Phase

All CHs are initially thought of as, for instance, provisional routers. The CHs are assessed using several membership functions. The three input variables for the fuzzy routing strategy are “Residual Energy”, “Distance to BS”, and “Nodes Connected After Clustering (NCC)”.

At first, all CHs are thought of as temporary routers. Additionally, the fuzzy system’s linguistic variables are used to evaluate the CHs. “Residual energy”, “Distance to the BS”, and “NCC” are the input parameters for the fuzzy technique for routing. The residual energy has the following linguistic variations: less, average, and more. The linguistic variables for “Distance to BS” are far, reachable, and near, whereas those for “NCC” are many, medium, and few.

The membership functions were chosen in accordance with [9] and the results of their own experimental work. All the membership functions (MFs) of the input parameters “Residual energy”, “NCC”, and “Distance to BS” are shown, respectively, in Figure 2, Figure 3 and Figure 4. The range of MFs for “Residual Energy” is from 0 to 1 because the maximum energy of the sensor node is assumed to be 1 J. Because there can be a maximum of 20 sensor nodes in the range in this network scenario, the range of MFs for “NCC” is from 0 to 20. The range of “Distance to BS” is presumed to be between 0 and 142 because the greatest distance to the BS is 141.42 m. Between the sink point (100, 100) and the sensor node at the area’s corner (0, 0), the maximum distance is computed.

The possibility that a CH will function as a router is taken into consideration for the first fuzzy output variable, Confidence. The MF values for confidence are smallest, smaller, small, small moderate, moderate, big moderate, big, bigger, and biggest. The membership functions for the output variable “Confidence” are shown in Figure 5. ‘Capacity’, the second output variable, also demonstrates the analogous nine linguistic variables, including compact, rather compact, less compact, compact bearable, bearable, huge bearable, huge, less huge, and more huge. The membership function for the ‘Capacity’ of the sensor node is shown in Figure 6. Each CH in a routing path is accommodated by carefully choosing the ranges of all membership functions using a variety of experimental evaluations. For the routing method, Table 1 defines a total of 27 fuzzy if-then rules. In this case, defuzzification is accomplished using the centroid technique, whereas fuzzification is accomplished using the Mamdani model [20].

### 3.2. Router Selection and Routing Path Formation Phase

An initial arrangement of each CH is based on its distance from the BS. Within their communication range, the CHs transmit ‘candidate’ messages. Only the CHs are able to receive “candidate” messages. ‘Confidence’ and ‘Capacity’ information is included.

The CHs that are quite remote from the BS cannot function as routers; hence, they are only regarded as CHs. The CHs can participate in the routing process as the BS’s distance reduces. Different numbers of candidate messages are sent to each CH. Following the ‘candidate’ communication, CH assesses its confidence and capability in relation to the candidates that have been sent. The receiving CH will send a “join” message to the candidate CH if the candidate CHs’ confidence and capacity are higher than those of the receiving CH. The potential CHs may now get several ‘join’ notifications. So, in accordance with their capacity, they send “accept” messages to their closer CHs. The CHs that sent the “accept” message to them also send their data to those CHs. The CHs that send the “accept” message are known as “routers”. Data is sent from CHs and routers to the routers that are on the routing path leading to the BS. Any sensor node sending data straight to the BS does not receive an “accept” message. As a result, the data from all the CHs is sent in this way to the BS. Table 2 displays the message summaries.

The complexity can be amounted according to the time required for sending messages (*t*) for one CH. The number of CHs selected in the worst case would be a predefined number *k* at each iteration. If *N* is the total number of iterations, then the worst case for complexity amounts to O(tkN). The flow chart of the proposed EERF is shown in Figure 7.

## 4. Discussion

The simulations were run on Windows 10 using MATLAB R2015. With the placement of 100 sensor nodes at random, the area under observation was considered to be 200 × 200 $m^{2}$. Each sensor node has an initial energy level of 1 J. The energy needed for intra-cluster activities is calculated as $E_{elec}=50$ nJ/bit, $\epsilon_{fs}=10\;\mathrm{pJ}/\mathrm{bit}$/$m^{2}$, and 10% of the data was aggregated.

This was compared to two well-known methodologies, EAUCF and DUCF, in order to assess the performance of the suggested methodology. In this work, the network’s overall energy consumption throughout each round, the number of active sensor nodes during a round, and the stability of the network were all evaluated during the simulations. The network’s stability is defined as the difference between the first sensor node to die (FND) and half of the sensor nodes having died (HND). The stable network has a larger difference.

Figure 8 displays the network’s sensor nodes’ cumulative energy use for each round. Performance-wise, the suggested method outperforms EAUCF and DUCF. The number of neighbor nodes is not taken into account as an input parameter in EAUCF. As a result, the network’s total load is not balanced. Although node degree is taken into account in DUCF, the direct communication between the CHs and BS was applied. Because of this, the CHs consume more energy and eventually perish. Both of these criteria were taken into account, and a routing algorithm was also suggested for use in the EERF. As a result, it uses less energy during each round.

The network’s overall number of active sensor nodes for each round is shown in Figure 9. It demonstrates that the line graph of the proposed approach up until the 1000th round displays the greatest values of all the algorithms that were compared. This is a result of the planned EERF’s lower energy consumption. EERF conserves energy and outlasts the other compared techniques across all rounds.

The FND and HND values are recorded by the network. Table 3 shows the network stability, FND, and HND values, as well as the difference between the two. It was found that the suggested approach significantly differs from FND and HND. In comparison to EAUCF and DUCF, it is, therefore, more stable.

The energy usage at each round is tracked because energy efficiency is the main goal of this endeavor. The amount of energy used during each loop is shown in Figure 10. For this parameter, it can be seen that EERF performs better than EAUCF and DUCF. When compared to EAUCF and DUCF, EERF uses a lot less energy. As the number of rounds in which the FND in the network and the HND in the network are noted, the relative energy consumption at each round is observed. When the number of rounds increases, the network is not stable and consumes a high amount of energy for transmission. So, the rate of death of sensor nodes is higher.

According to all the results, the proposed EERF algorithm outperforms EAUCF and DUCF and generates energy-efficient communications in the routing. Due to the energy-efficient communications and routing path, the network stability increases. Consequently, a better network lifespan is achieved.

## 5. Conclusions

As a result of the battery-powered sensor nodes in WSNs, an efficient level of energy consumption is essential. In this study, fuzzy logic is used to propose a routing algorithm that is energy-efficient. The outputs from the fuzzy inference system are used to detect the appropriate routers and the routing path. The proposed EERF is compared to the current EAUCF and DUCF under several assessment criteria, including total network energy consumption, active sensor nodes, network stability, and energy consumption at each round, in order to assess how well it performs. It was discovered that EERF outperforms EAUCF and DUCF as a result of the fuzzy system’s addition of the parameter “NCC” and the efficient generation of routing paths. Furthermore, EERF can be improved with different clustering techniques and considering various computational parameters in terms of the sensor node’s capacity to transfer data and the node degree for the selection of CHs.

## Figures and Tables

**Figure 1 sensors-23-08074-f001:**
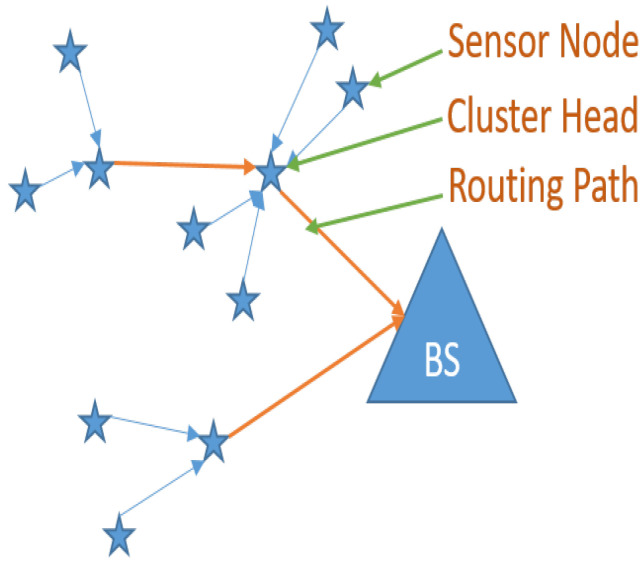
Wireless sensor network with clusters.

**Figure 2 sensors-23-08074-f002:**
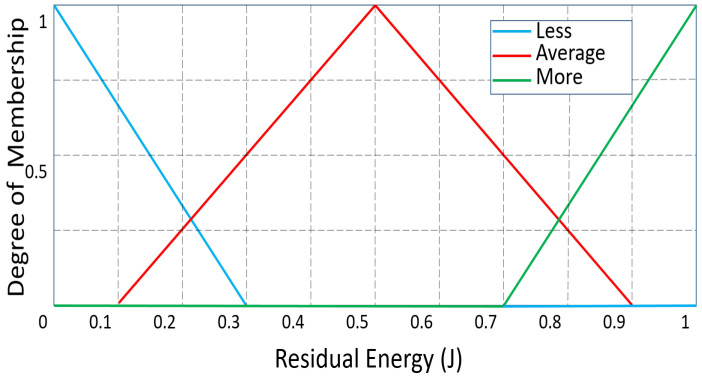
‘Residual Energy’ as an input variable: degree of membership.

**Figure 3 sensors-23-08074-f003:**
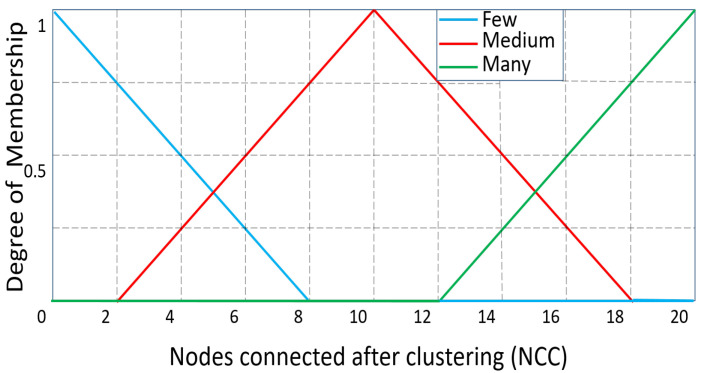
‘NCC’ as an input variable: degree of membership.

**Figure 4 sensors-23-08074-f004:**
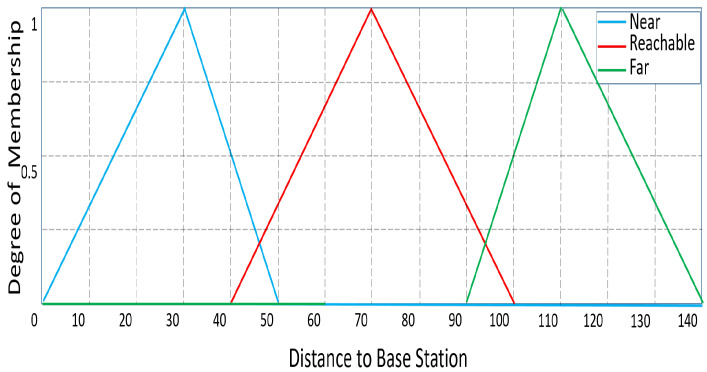
‘Distance to BS’ as an input variable: degree of membership.

**Figure 5 sensors-23-08074-f005:**
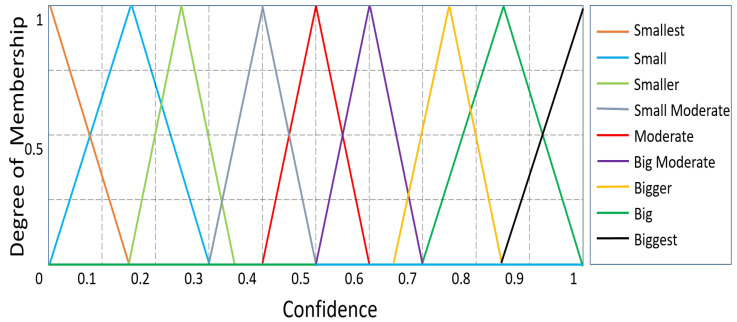
’Confidence’ as an output variable: degree of membership.

**Figure 6 sensors-23-08074-f006:**
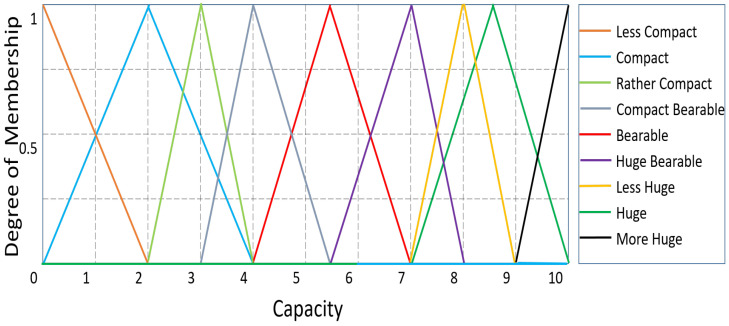
’Capacity’ as an output variable: degree of membership.

**Figure 7 sensors-23-08074-f007:**
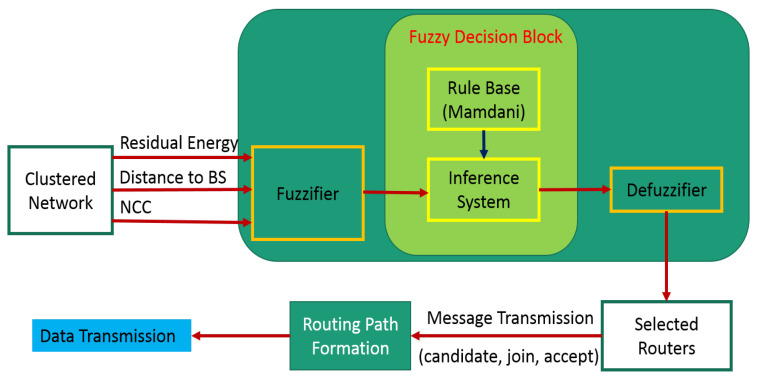
Flowchart of proposed EERF.

**Figure 8 sensors-23-08074-f008:**
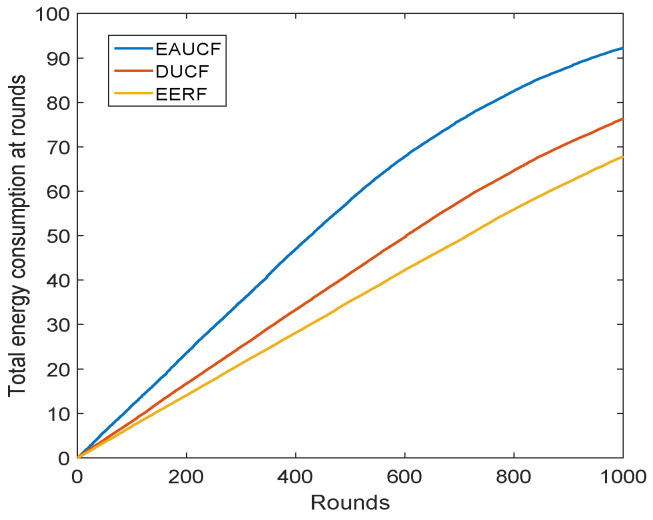
Total energy consumption per round in Joules.

**Figure 9 sensors-23-08074-f009:**
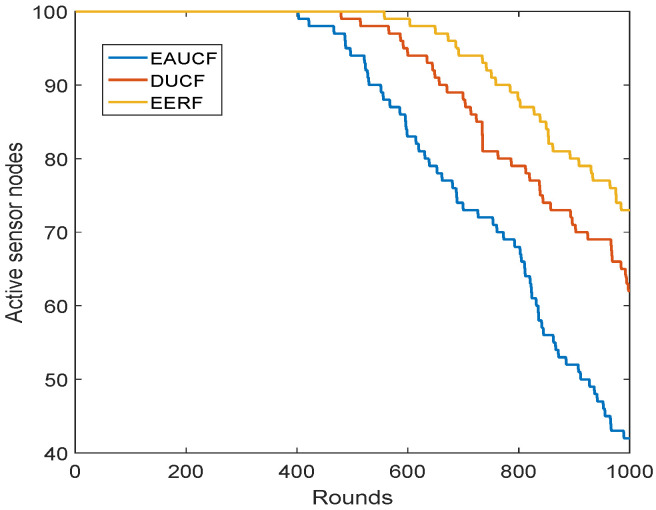
Active sensor nodes per round.

**Figure 10 sensors-23-08074-f010:**
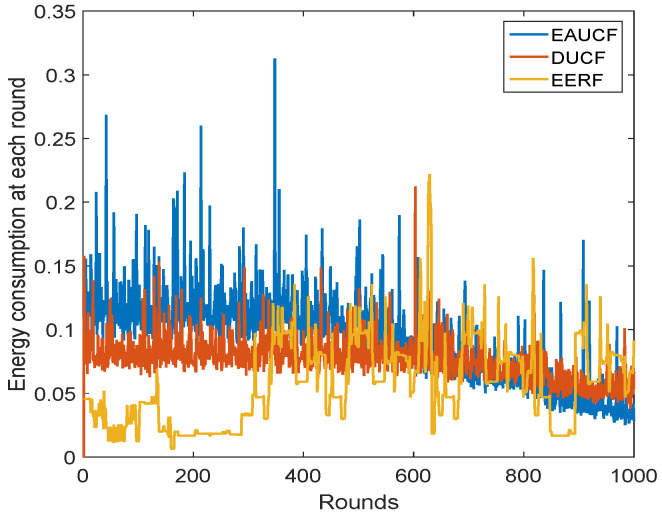
Energy consumption at each round in Joules.

**Table 1 sensors-23-08074-t001:** Fuzzy if-then rules for routing in WSN.

Sr. No	Input Variables			Output Variables	
	Residual Energy	Distance to BS	NCC	Confidence	Capacity
1	Less	Far	Many	Smallest	Compact
2	Less	Far	Medium	Smaller	Rather Compact
3	Less	Far	Few	Small	Less Compact
4	Less	Reachable	Many	Small Moderate	Compact
5	Less	Reachable	Medium	Moderate	Rather Compact
6	Less	Reachable	Few	Big Moderate	Less Compact
7	Less	Near	Many	Big	Compact
8	Less	Near	Medium	Bigger	Rather Compact
9	Less	Near	Few	Biggest	Less Compact
10	Average	Far	Many	Smallest	Compact Bearable
11	Average	Far	Medium	Smaller	Bearable
12	Average	Far	Few	Small	Huge Bearable
13	Average	Reachable	Many	Small Moderate	Compact Bearable
14	Average	Reachable	Medium	Moderate	Bearable
15	Average	Reachable	Few	Big Moderate	Huge Bearable
16	Average	Near	Many	Big	Compact Bearable
17	Average	Near	Medium	Bigger	Bearable
18	Average	Near	Few	Biggest	Huge Bearable
19	More	Far	Many	Smallest	Huge
20	More	Far	Medium	Smaller	Less Huge
21	More	Far	Few	Small	More Huge
22	More	Reachable	Many	Small Moderate	Huge
23	More	Reachable	Medium	Moderate	Less Huge
24	More	Reachable	Few	Big Moderate	More Huge
25	More	Near	Many	Big	Huge
26	More	Near	Medium	Bigger	Less Huge
27	More	Near	Few	Biggest	More Huge

**Table 2 sensors-23-08074-t002:** Message descriptions.

Sr. No	Message	Description
1	candidate	Initially, each CH broadcasts candidate in the communication range R.
2	join	If Confidence and Capacity of candidate node is higher then, send accept message.
3	accept	If Capacity is not exceeding, then send accept to nearer CHs.

**Table 3 sensors-23-08074-t003:** Comparison of FND, HND, and the stability of the network.

Algorithm	FND	HND	Stability Check
EAUCF	403	928	525
DUCF	480	1012	532
EERF	**558**	**1161**	**603**

## Data Availability

Not applicable.

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
