# Peer review of "An Energy-Efficient Routing Algorithm for WSNs Using Fuzzy Logic"

_sensors, 2023, doi:10.3390/s23198074_

Round 1

Reviewer 1 Report

The authors propose an Energy efficient Routing algorithm using fuzzy logic (EERF) for Wireless Sensor Networks (WSN) applications. This algorithm has a novel approach in order to create an energy-efficient data transfer technique. The authors used as input of the EERF algorithm the remaining energy, the distance between the sensor node and the base station, along with the total number of connected sensor nodes. Besides, they contrasted the proposed EERF with the current systems like LEACH and DUCF algorithm in terms of evaluation criteria including energy consumption, the number of active sensor nodes for each round in the network as well as network stability. LEACH and DUCF are outperformed by EERF. In my view point, the paper has a novel approach and it is a relevant topic Therefore, I suggest that the following major revisions be considered for acceptance.

1.- It mentions that the proposed EERF is compared with the current systems such as LEACH and DUCF algorithm in terms of evaluation criteria including energy consumption, number of active sensor nodes for each round in the network as well as network stability. However, I do not see any focus or comparison on this in the discussion or results section. For instance, just LEACH is mentioned seven times in the whole manuscript.

2.- What analysis of the results leads to the conclusion that LEACH and DUCF are outperformed by EERF?

4.-The complexity analysis of the EERF algorithm proposed and the flux diagram should be included.

5.- Some acronyms are not defined (for instance: in the abstract “LEACH and DUCF” ), please do so.

6.- Please, some acronyms are defined more than once (by example “Nodes connected after clustering (NCC)”), check it.

7.- In figures 7, 8, and 9, what are the units of the y-axes?

8.- When compared to EAUCF and DUCF, the EERF uses much less energy at 600 rounds, but after,  it possible see that the EAUDC and DUCT decrease the energy consumption until equal to the EERF proposed. What happens at more than 100 rounds? it is possible to say that EERF has energy consumption minor at short rounds or initial rounds?

Minor editing

Author Response

Reviewer 1 Comments

In my view point, the paper has a novel approach and it is a relevant topic Therefore, I suggest that the following major revisions be considered for acceptance.

Author’s Response

Authors would like to sincerely thank the reviewer 1 for providing the opportunity for major revision and also providing valuable comments for improving the manuscript. The suggestions are incorporated in the revised manuscript.

S. No

Editor’s Comment

Author’s Response

1.

It mentions that the proposed EERF is compared with the current systems such as LEACH and DUCF algorithm in terms of evaluation criteria including energy consumption, number of active sensor nodes for each round in the network as well as network stability. However, I do not see any focus or comparison on this in the discussion or results section. For instance, just LEACH is mentioned seven times in the whole manuscript.

There was a typo mistake in the abstract. The proposed algorithm is compared with EAUCF and DUCF algorithm. The detailed comparison of the proposed algorithm with EAUCF and DUCF has been incorporated in the revised manuscript.

(Highlighted in Page Nos. 1 and 8-9)

2.

What analysis of the results leads to the conclusion that LEACH and DUCF are outperformed by EERF?

The least energy consumption of EERF algorithm and best network stability are the main reasons which leads to the conclusion that EAUCF and DUCF are outperformed by EERF.

3.

The complexity analysis of the EERF algorithm proposed and the flux diagram should be included.

As per the suggestion, the complexity analysis of the EERF algorithm and the flow chart is included in the revised manuscript.

(Highlighted in Page Nos. 7-8)

4.

Some acronyms are not defined (for instance: in the abstract “LEACH and DUCF” ), please do so.

All the acronyms are defined in the revised manuscript.

(Highlighted in Page Nos. 1-2)

5.

Please, some acronyms are defined more than once (by example “Nodes connected after clustering (NCC)”), check it.

The acronyms are defined only once in the revised manuscript.

(Highlighted in Page Nos. 5 and 10)

6.

In figures 7, 8, and 9, what are the units of the y-axes?

In Figure 8 (previous Fig 7) the unit of y-axis is in Joules. The Figure 9 (previous Fig 8), shows the number of active sensor nodes per round.  In Figure 9 (previous Fig 8), the unit at y axis is in Joules.

(Highlighted in Page Nos. 8-9)

7.

When compared to EAUCF and DUCF, the EERF uses much less energy at 600 rounds, but after,  it possible see that the EAUDC and DUCT decrease the energy consumption until equal to the EERF proposed. What happens at more than 100 rounds? it is possible to say that EERF has energy consumption minor at short rounds or initial rounds?

After 400-450 rounds the nodes stop working due to insufficient energy in EAUCF and DUCF. Therefore, total energy consumption at each round is reduced. This even happens for the proposed EERF but after 558 rounds, which shows good stability of the network.

Author Response

Reviewer 2 Comments

This study addresses a routing algorithm for WSN. The proposed method – EERF was simulated using MATLAB, compared with the other methods.

There are a number of issues:

Author’s Response

Authors would like to sincerely thank the reviewer 2 for providing valuable comments for improving the manuscript. All the issues are addressed below.

S. No

Reviewer’s Comment

Author’s Response

1.

The introduction section is too brief to cover the research topic. It should include a comprehensive description for the studied problem.

As per the suggestion, the introduction section has been improved, including a comprehensive description of the problem statement.

(Highlighted in Page No. 2)

2.

For the proposed method, are there any mathematical equations to describe the algorithm/method?

There are no mathematical equations used in the manuscript except the energy model.

3.

The discussion section only produces a limited simulation result. The advantage and disadvantages of the method need more discussion.

As suggested, the discussion section has been improved with detailed explanation of simulation results. Also, the advantages and the disadvantages of the proposed method are discussed in the revised manuscript.

(Highlighted in Page No. 9)

4.

Figure 9’s explanation appears insufficient. It shows for smaller rounds, the energy consumption is lower. When the rounds approach large values, it doesn’t

The detailed explanation of Figure 9 is included in the manuscript. As we have noted the number of rounds where the first node dies in the network and half of the node dies in the network, the relative energy consumption at each round is observed. When the number of rounds will increase, the network will not be stable and consume a high amount of energy for transmission. So the rate of death of sensor nodes will be higher.

(Highlighted in Page No. 9)

5.

Only compared with simulated results, anything compared with realistic experimental measurements?

The results are noted from the simulator whereas no real time experimental results has been measured. However, when the algorithm will be applied to the real time sensor notes, there might be some minor deviations in the results due to hardware computations.

6.

Conclusion – too brief. Future work?

Conclusion has been improved, including the analysis of the results. The future work is also included in the revised manuscript.

(Highlighted in Page No. 10)

7.

References are few- not sufficient to cover the research topic

The reference section has been improved, including the latest work in order to cover the research topic.

(Highlighted in Page No. 11)

8.

Minor spelling: in Abstract – given; affiliation: what is Goa?

The typo in abstract is corrected. Goa is a state in India.

(Highlighted in Page No. 1)

Reviewer 3 Report

The manuscript is very well written and clear. Some minor issues were found regarding writing:

1) Whether in Section 1 or 2, it is suggested to clearly state the problem of decreasing energy consumption and to highlight the differences (advantages) of the proposed solution from other approaches.

2) The equations must be referenced with parenthesis, for instance, Equation (1) instead of Equation 1. The authors might use the  \eqref command if they are using Latex. 

3) In Subsection 2.1 Paragraphs 1 and 2, the authors should write in math mode "p" and "T" for the predetermined number of rounds and the Threshold value, respectively, since they are variables that will be used later in the equations.

4) In Subsection 2.2 Paragraph 1, the authors should also write in math mode "l" and "d" of bits length and meters, respectively. 

Regarding technical aspects, the following questions arose:

5) The membership functions "Near" and "Far" are triangular, leading to low membership values before 20 meters and after 110 meters, respectively. How correct weighting and mapping are guaranteed in such ranges? For instance, a 1 m distance might not be considered "Near" enough due to the low membership value. Would trapezoid membership functions yield a better weighting?

6) It would be interesting to have graphical information about the mapping from the IF-THEN rules, like the surfaces, so that the fuzzy system could be analyzed better by the readers. It could give better information about the weighting as well.

The paper is very well written. Minor corrections are required.

Author Response

Reviewer 3 Comments

The manuscript is very well written and clear. Some minor issues were found regarding writing:

Author’s Response

Authors would like to sincerely thank the reviewer 3 for providing the opportunity for minor revision and also providing valuable comments for improving the manuscript. All the issues are addressed below.

S. No

Reviewer’s Comment

Author’s Response

1.

Whether in Section 1 or 2, it is suggested to clearly state the problem of decreasing energy consumption and to highlight the differences (advantages) of the proposed solution from other approaches.

The problem of decreasing energy consumption is stated in Section 1. Also, the advantages of proposed solution over existing approaches are highlighted in Section 1.

(Highlighted in Page No. 2)

2.

The equations must be referenced with parenthesis, for instance, Equation (1) instead of Equation 1. The authors might use the  \eqref command if they are using Latex.

The equations are referred as per the suggestion.

(Highlighted in Page No. 4)

3.

In Subsection 2.1 Paragraphs 1 and 2, the authors should write in math mode "p" and "T" for the predetermined number of rounds and the Threshold value, respectively, since they are variables that will be used later in the equations.

The suggestions are incorporated in the revised manuscript.

(Highlighted in Page Nos. 2-3)

4.

In Subsection 2.2 Paragraph 1, the authors should also write in math mode "l" and "d" of bits length and meters, respectively.

The suggestions are incorporated in the revised manuscript.

(Highlighted in Page No. 4)

5.

The membership functions "Near" and "Far" are triangular, leading to low membership values before 20 meters and after 110 meters, respectively. How correct weighting and mapping are guaranteed in such ranges? For instance, a 1 m distance might not be considered "Near" enough due to the low membership value. Would trapezoid membership functions yield a better weighting?

The sensor node within 1 meter or after 110 meters distance will be considered as near and far respectively to the CH. Both the sensor nodes carry membership value as only near and far functions will be considered. In this case, as the different ranges are decided for each membership function the trapezoidal membership function may not guarantee the good results due to same range of the distance to BS.

6.

It would be interesting to have graphical information about the mapping from the IF-THEN rules, like the surfaces, so that the fuzzy system could be analyzed better by the readers. It could give better information about the weighting as well.

As if- then rules are easy to understand, and are very popular, we have preferred to include them in the manuscript.

Round 2

Reviewer 1 Report

All changes have been made by the authors.

Minor editing